# Tetrahelical structural family adopted by AGCGA-rich regulatory DNA regions

Vojč Kocman[1] & Janez Plavec[1,2,3]

Here we describe AGCGA-quadruplexes, an unexpected addition to the well-known tetrahelical families, G-quadruplexes and i-motifs, that have been a focus of intense research due to their potential biological impact in G- and C-rich DNA regions, respectively. High-resolution structures determined by solution-state nuclear magnetic resonance (NMR) spectroscopy demonstrate that AGCGA-quadruplexes comprise four 5′-AGCGA-3′ tracts and are stabilized by G-A and G-C base pairs forming GAGA- and GCGC-quartets, respectively. Residues in the core of the structure are connected with edge-type loops. Sequences of alternating 5′-AGCGA-3′ and 5′-GGG-3′ repeats could be expected to form G-quadruplexes, but are shown herein to form AGCGA-quadruplexes instead. Unique structural features of AGCGA-quadruplexes together with lower sensitivity to cation and pH variation imply their potential biological relevance in regulatory regions of genes responsible for basic cellular processes that are related to neurological disorders, cancer and abnormalities in bone and cartilage development.

[1] Slovenian NMR Centre, National Institute of Chemistry, SI-1000 Ljubljana, Slovenia. [2] EN-FIST Centre of Excellence, SI-1000 Ljubljana, Slovenia. [3] Faculty of Chemistry and Chemical Technology, University of Ljubljana, SI-1000 Ljubljana, Slovenia. Correspondence and requests for materials should be addressed to J.P. (email: janez.plavec@ki.si).

The double helix, the most common secondary structure of DNA, protects nucleotides from damage that is crucial for safeguarding genetic information in the form of nucleotide sequence. However, when DNA is involved in processes such as replication and transcription, its double helical structure is partially unwound into two strands. Switching to single strands under certain conditions leads to formation of higher-order DNA structures that can interfere or even stop replication possibly resulting in harmful mutations[1]. Higher-order DNA structures are processed by various enzymes such as DNA helicases. Two families of higher-order tetrahelical DNA structures called G-quadruplexes and i-motifs are formed by guanine- and cytosine-rich oligonucleotides, respectively[2–4]. G-quadruplexes are tetrahelical structures consisting of stacked G-quartets, cyclic arrangements of guanine residues held together by hydrogen bonds in Hoogsteen geometry organized around a central cation[5–9]. Bioinformatic analysis together with other approaches such as polymerase stop assay revealed that nucleotide sequences capable of forming G-quadruplexes are highly populated in regions implicated in essential cellular processes such as initiation of DNA replication, telomere maintenance, recombination in immune evasion and response, control of gene expression and genetic and epigenetic instability[10,11]. G-quadruplexes have been visualized in human cells with the use of immunofluorescence[12]. On the other hand, i-motifs consist of intercalated cytosine-cytosine[+] base pairs joining C-rich strands[4]. Initially, i-motifs were thought to be stable only at acidic pH conditions that impeded more extensive biological investigations. Lately, it has been shown that they can also be stable at near physiological pH values[13–16]. Our recent finding of an unusual four-stranded DNA structure[17] stimulated us to explore whether G- and A-rich repeat segments of DNA can adopt tetrahelical structures different from G-quadruplexes and i-motifs. We were encouraged by polymerase stop assay experiments that identified stalling at over 100,000 G-rich sites in the human genome (out of 700,000 total G-rich sites that caused polymerase stalling) that did not adhere to the consensus G-quadruplex folding motif of 5′-GGG(N$_{1–7}$)GGG(N$_{1–7}$)GGG(N$_{1–7}$)GGG-3′ and could not simply be explained by G-quadruplexes containing long loops (>7 nucleotides) or single-nucleotide bulges[11]. Alternatively, G-rich sites could form complex G-quadruplex topologies containing only two G-quartets or extended bulges, possibly even in combination with long loops[11].

We set out to examine a tempting hypothesis that the identified G- and A-rich sites could form yet unknown structures similar to tetrahelical topologies adopted by oligonucleotides that contain two and four 5′-GGGAGCG-3′ repeats found in the regulatory region of the *PLEKHG3* human gene related to autism[17]. In contrast to the expected G-quartet-based topologies adopted by 5′-GGG-3′ repeats, both unimolecular structure and its dimeric analogue are stabilized by G-C, G-A and G-G base pairs that interact to form a unique structure[17]. Our bioinformatic analysis revealed 146 sequences with at least 4 5′-AGCGA-3′ repeats, 46 of which are found in regulatory regions of 38 human genes implicated in basic cellular processes, neurological disorders, cancer and abnormalities in bone and cartilage development. Herein we are the first, to the best of our knowledge, to describe topologies and high-resolution structures of AGCGA-quadruplexes, a tetrahelical family adopted by oligonucleotides that contain 5′-AGCGA-3′ repeats. AGCGA-quadruplexes are characterized by tetrahelical cores of 5′-AGCGA-3′ repeats connected with edge-type loops of various lengths mostly stabilized by G-G base pairs in N1-carbonyl symmetric geometry. AGCGA-cores can form GAGA-quartets, major and minor groove GCGC-quartets or instead have G-A base pairs stacked on G-G base pairs from loop regions. Surprisingly, AGCGA-quadruplexes are formed even when 5′-AGCGA-3′ repeats are separated by 5′-GGG-3′ tracts and corresponding sequences adhere to the G-quadruplex consensus sequence. In addition, we provide a rationale that formation of tetrahelical structures is guided by a specific folding mechanism involving pre-folded duplexes and demonstrate the role of cations that not only neutralize the negatively charged sugar-phosphate backbone of DNA, but specifically stabilize different AGCGA-quadruplex structures.

## Results

**Unique tetrahelical fold stabilized by GAGA-quartets.** At the outset, based on the oligonucleotide sequence, 15-mer 5′-d(GCGAGGGAGCGAGGG)-3′, VK34, was expected, to form either a G-quadruplex with a GCGC-quartet sandwiched between two G-quartets or a dimeric structure similar to the one adopted by 5′-d(GGGAGCGAGGGAGCG)-3′, VK1 (ref. 17). Expected topologies were based on uncommon base pair geometries exemplified in Fig. 1 that were indeed observed in the subsequently determined structures.

One-dimensional proton nuclear magnetic resonance (1D $^1$H NMR) spectrum of VK34 reveals a set of well-resolved resonances that suggest a single fold (Fig. 2 and Supplementary Fig. 1). Native PAGE (polyacrylamide gel electrophoresis) gel mobilities, analytical ultracentrifugation and translational

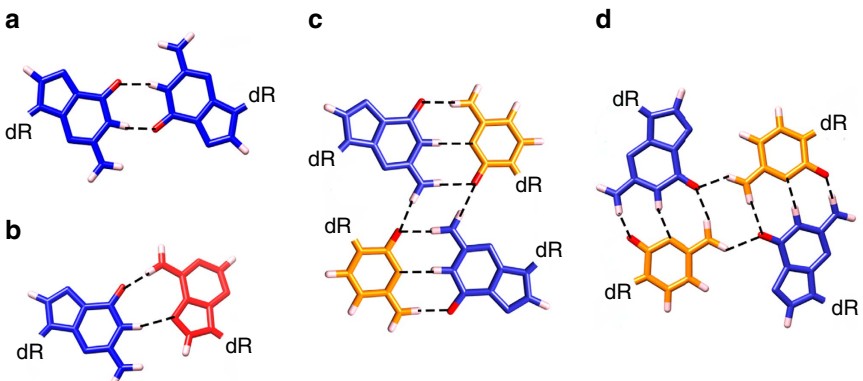

**Figure 1 | G-G and G-A base pairs as well as minor and major groove GCGC-quartets.** (**a**) G-G base pair in N1-carbonyl symmetric geometry. (**b**) G-A base pair in N1-N7, carbonyl-amino geometry. (**c**) Minor groove GCGC-quartet. (**d**) Major groove GCGC-quartet. Coordinates are taken from high-resolution structures of VK34 (PDB ID: 5M1L and 5M2L). Deoxyribose moieties are represented by dR. The guanine residues are coloured blue, adenine red and cytosine orange. Hydrogen and O6 atoms are coloured white and red, respectively.

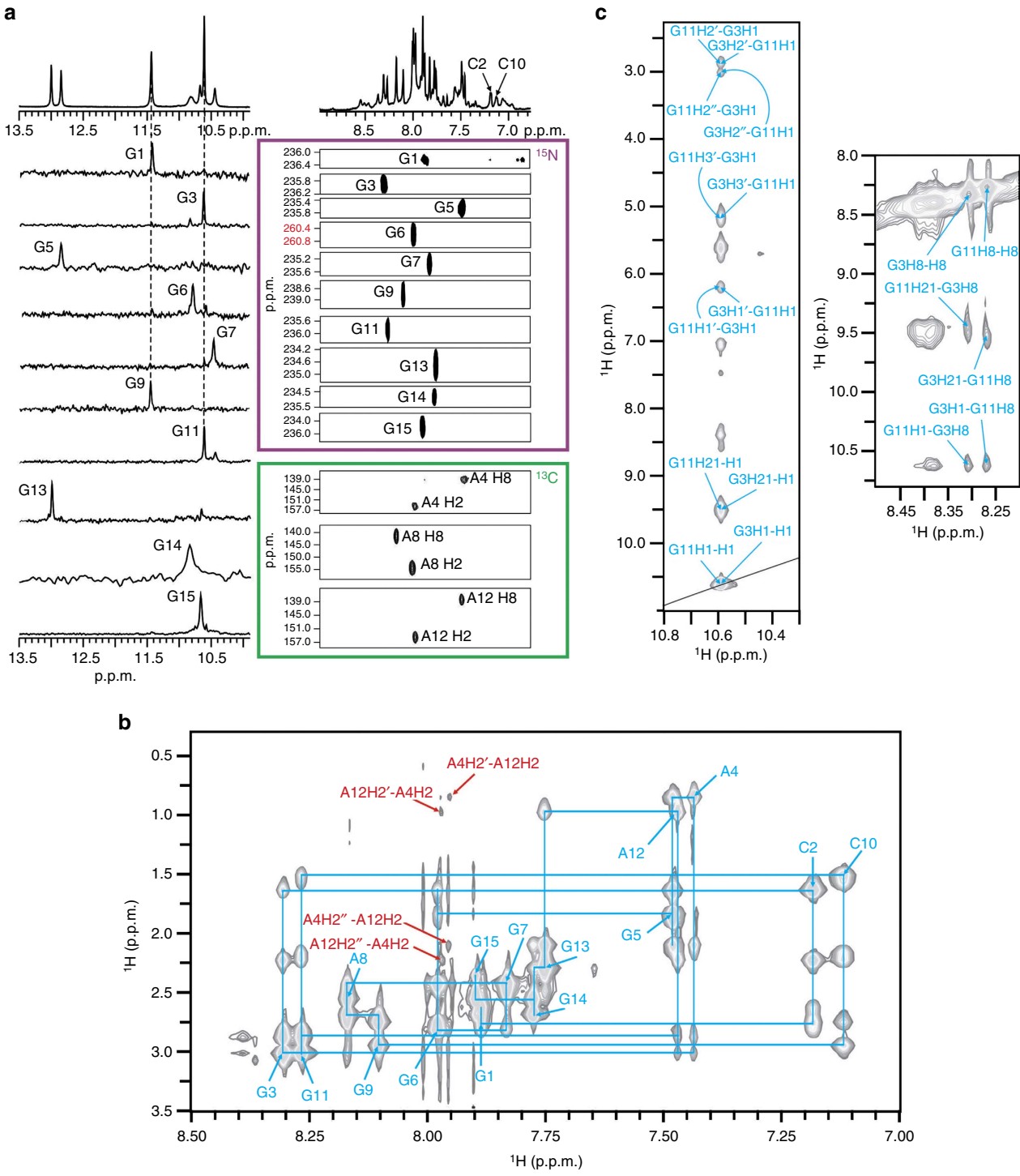

**Figure 2 | Unambiguous assignment and crucial regions of NOESY spectrum of VK34. (a)** Imino region of 1D $^{1}$H NMR spectrum above 1D $^{15}$N-edited HSQC spectra and aromatic region of 1D $^{1}$H NMR spectrum above 2D $^{15}$N- and $^{13}$C-edited heteronuclear multiple-quantum correlation spectroscopy (HMQC) spectra. The HSQC and HMQC spectra were acquired on partially (10%) residue-specifically $^{15}$N- and $^{13}$C-labelled oligonucleotides. Assignment of H1 and H8 proton resonances is indicated next to the 1D signals and 2D cross-peaks, respectively. Assignment of H2 and H8 adenine proton resonances is indicated next to the cross-peaks. The dashed lines signify the assignment of G1, G9 and G3, G11 pairs whose H1 protons are nearly isochronous. The H6 proton resonances of C2 and C10 residues are indicated in the aromatic region of 1D $^{1}$H NMR spectrum. The red colour in the 2D $^{15}$N-edited HMQC spectrum indicates the unusual downfield chemical shift of G6 N7 atom. **(b)** H2'/H2''-aromatic region of NOESY spectrum (mixing time of 200 ms) of VK34 at 0.8 mM oligonucleotide concentration per strand, 100 mM LiCl concentration, pH 6.0 and 0 °C. The sequential walk is depicted in blue with H8 proton resonances indicated next to intraresidual cross-peaks. NOE contacts observed between A4 and A12 are depicted in red. **(c)** Regions of NOESY spectrum that show contacts between G3 and G11 residues.

diffusion experiments ($D_t = 0.6 \times 10^{-10}$ m$^2$ s$^{-1}$) are in agreement with dimeric nature of the fold (Supplementary Figs 2 and 3). A single set of NMR signals is in accordance with its symmetric structure. The guanine H1 and H8 as well as adenine H2 and H8 proton resonances were unambiguously assigned with the help of $^{15}$N- and $^{13}$C-edited spectra acquired on partially (10%) residue-specifically $^{15}$N- and $^{13}$C-labelled VK34 (Fig. 2a).

A structurally independent approach enabled us to assign chemical shifts of a majority of proton resonances including some of H4', H5' and H5" resonances, consistent with antiparallel, dimeric, head-to-head topology. The 458 nuclear Overhauser enhancement (NOE)-derived distance restraints together with 66 torsion angle and 24 hydrogen bond restraints were used to calculate the high-resolution structure of VK34 (PDB ID: 5M1L, Supplementary Table 1 and Fig. 3). In the dimeric structure, all G, C and A residues from one VK34 molecule, except A8, are involved in hydrogen bonding with residues from the other VK34 molecule (Fig. 3a). Even though the four strands are orientated in an antiparallel fashion, all residues assume *anti* orientations across glycosidic bonds. G3-A12-G3-A12 and G11-A4-G11-A4 quartets, which are stacked on each other, are located in the centre of the structure. GAGA-quartets consist of a couple of G-A base pairs in N1–N7, carbonyl-amino geometry, that are connected within a plane by N7, amino hydrogen bonds (Figs 1b and 3b). Our NMR data analysis shows that formation of GAGA-quartets does not follow the folding rules for G-quartets, where *syn* and *anti* guanine orientations are altered simultaneously within a G-quartet and along antiparallel strands. In comparison with G-quartet stacking modes observed in G-quadruplexes, GAGA-quartets exhibit stacking interactions characterized by a large helical twist (Fig. 3c). This enables interstrand G3 and G11 as well as intrastrand A4 and A12 stacking interactions with their pyrimidine moieties. In turn, sequential stacking interactions of G3 with A4 and G11 with A12 residues are reduced. The distinctive stacking mode is supported by a unique NOE cross-peak fingerprint, where each H1(guanine) and H2(adenine) proton exhibits NOE connectivities with sugar and H8 protons of a stacked partner in 6|6-ring stacking geometry (Fig. 2b,c and Supplementary Fig. 4). Two adjacent

G5-C10 base pairs face each other with their minor groove sides and connect in a way that guanine amino protons form hydrogen bonds with the cytosine O2 atom involved in a nearby Watson–Crick base pair (Figs 1c and 3d). This 'minor groove' G5-C10-G5-C10 quartet is stacked on a G11-A4-G11-A4 quartet (Fig. 3a). Minor groove GCGC-quartets are rare structural elements since they are characterized by two extremely narrow grooves, located opposite to each other, that cause major disruptions in progression of wider grooves such as the ones defined by stacked G-quartets. The stacked G3-A12-G3-A12 and G11-A4-G11-A4 quartets define four medium grooves. Two of them experience a pronounced narrowing between intrastrand G5 and C10 residues of the G5-C10-G5-C10 quartet (Supplementary Fig. 5). The strands defining narrow grooves are positioned close enough to allow formation of G9-G9, G6-G6 and G7-G7 base pairs in N1-carbonyl symmetric geometry (Fig. 1a). The G9-G9 and G6-G6 base pairs are arranged in a crisscross topology, whereas the G7-G7 base pair is stacked on the G6-G6 base pair (Fig. 3d). A8 residues facilitate a change in strand directionality leading to an antiparallel overall topology of the VK34 dimer. On the opposite side of the structure, two C2-G13 base pairs that exhibit a slight buckle are positioned below the G3-A12-G3-A12 quartet (Fig. 3a,e). Since C2-G13 base pairs do not associate into a GCGC-quartet, narrowing of the groove between two intramolecular strands is not as pronounced as for the G5-C10-G5-C10 quartet. The strands are still close enough to facilitate formation of G1-G1, G14-G14 and G15-G15 base pairs in N1-carbonyl symmetric geometry. The G1-G1 and G14-G14 base pairs are arranged in a crisscross topology and the latter is stacked on the G15-G15 base pair.

**GAGA-quartets as a versatile platform for tetramers**. In addition to folding into a dimeric structure, VK34 demonstrates a concentration- and time-dependent transition to a second fold characterized by a new set of signals in 1D $^1$H NMR spectra. At VK34 oligonucleotide concentrations close to 0.8 mM, complete transition to a new fold takes a couple of weeks. At concentrations close to 2.0 mM, the second fold is dominant

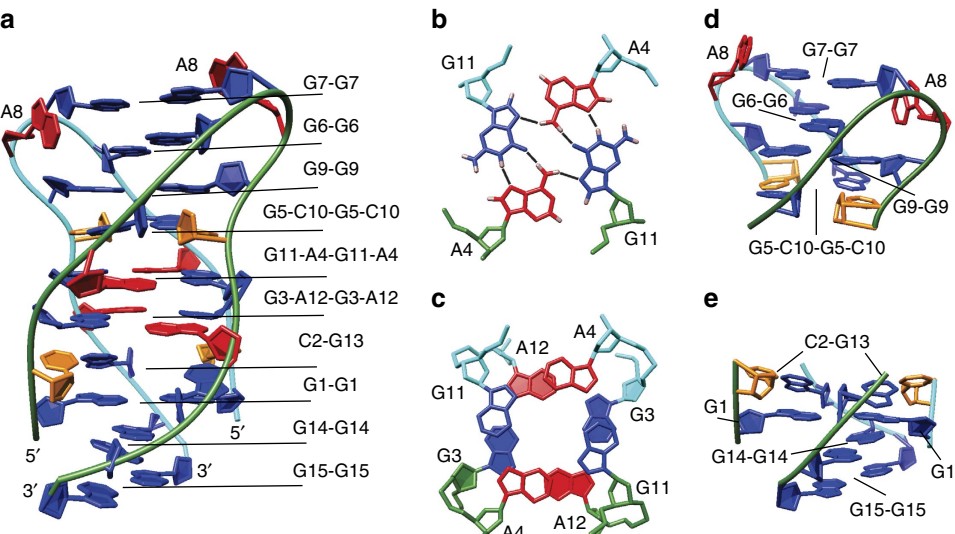

**Figure 3 | Structure of VK34 dimer.** (**a**) Representation of the lowest-energy overall structure. (**b**) Top view of the G11-A4-G11-A4 quartet with marked hydrogen bonds. (**c**) Top view of stacked G3-A12-G3-A12 and G11-A4-G11-A4 quartets with a helical twist of 25° depicted with filled and unfilled residues, respectively. (**d**) Side view of the G5-C10-G5-C10 quartet and G6-G6, G7-G7 and G9-G9 base pairs. (**e**) Side view of two buckled C2-G13 base pairs and G1-G1, G14-G14 and G15-G15 base pairs. The guanine residues are coloured blue, adenine red and cytosine orange. The two strands are coloured green and cyan.

immediately after purification. Additionally, the same fold was achieved rapidly by titrating NaCl, KCl or NH$_4$Cl solutions into VK34 samples at oligonucleotide concentrations higher than 0.1 mM per strand (Supplementary Fig. 6). Native PAGE gel mobilities in the presence of 200 mM NaCl revealed that VK34 adopts a tetrameric fold that was supported by analytical ultracentrifugation and NMR translational diffusion experiments ($D_t = 0.3 \times 10^{-10}$ m$^2$ s$^{-1}$, Supplementary Figs 3 and 7). The guanine H1 proton resonances of the tetrameric fold were unambiguously assigned with the use of $^{15}$N-edited heteronuclear single-quantum coherence (HSQC) spectra acquired on partially (10%) residue-specifically $^{15}$N-labelled VK34 (Supplementary Fig. 8). With a simple comparison of nuclear Overhauser enhancement spectroscopy (NOESY) spectra of dimeric and tetrameric VK34 folds, we were able to assign H8 and sugar proton resonances of G1, G3, A4, G11, A12, G13, G14 and G15 residues, since these residues showed very similar NOESY fingerprints in both folds (Fig. 4a and Supplementary Fig. 9). H8 proton resonances of G5, G6, G7 and G9 residues were assigned unequivocally through selective elimination of cross-peaks in NOESY spectra upon complete (100%) D8 residue-specific deuteration (Supplementary Fig. 10). The adenine H2 as well as

cytosine H6 resonances were assigned from the cross-peaks that remained in the H2′/H2″-aromatic region when NOESY spectrum was recorded for the sample of VK34 with all G and A completely (100%) D8 labelled (Supplementary Fig. 11). In addition to familiar NOESY patterns observed for G1-A4 and G11-G15 segments, the G5-G9 region displayed numerous NOE contacts indicative of GAGA-, GCGC- and GGGG-quartets that enabled us to calculate a high-resolution tetrameric structure of VK34 with the use of 788 NOE-derived distance restraints together with 144 torsion angle and 88 hydrogen bond restraints (PDB ID: 5M2L, Fig. 5 and Supplementary Table 2).

The VK34 tetramer exhibits alternating strand directionalities (Fig. 5a). Its structure could be dissected into top, central and bottom parts. Each part contains a pair of stacked GAGA-quartets, characterized by a NOESY cross-peak fingerprint between stacked G and A residues also observed for stacked GAGA-quartets in the VK34 dimer (Fig. 4). Two stacked G7-A8-G7-A8 quartets are positioned in the centre of the structure and exhibit interchanging narrow and wide grooves characterized by increased G7-G7 and decreased A8-A8 overlap of their pyrimidine moieties, respectively (Supplementary Fig. 12). The G7-A8-G7-A8 quartets exhibit three perpendicular C$_2$ rotational

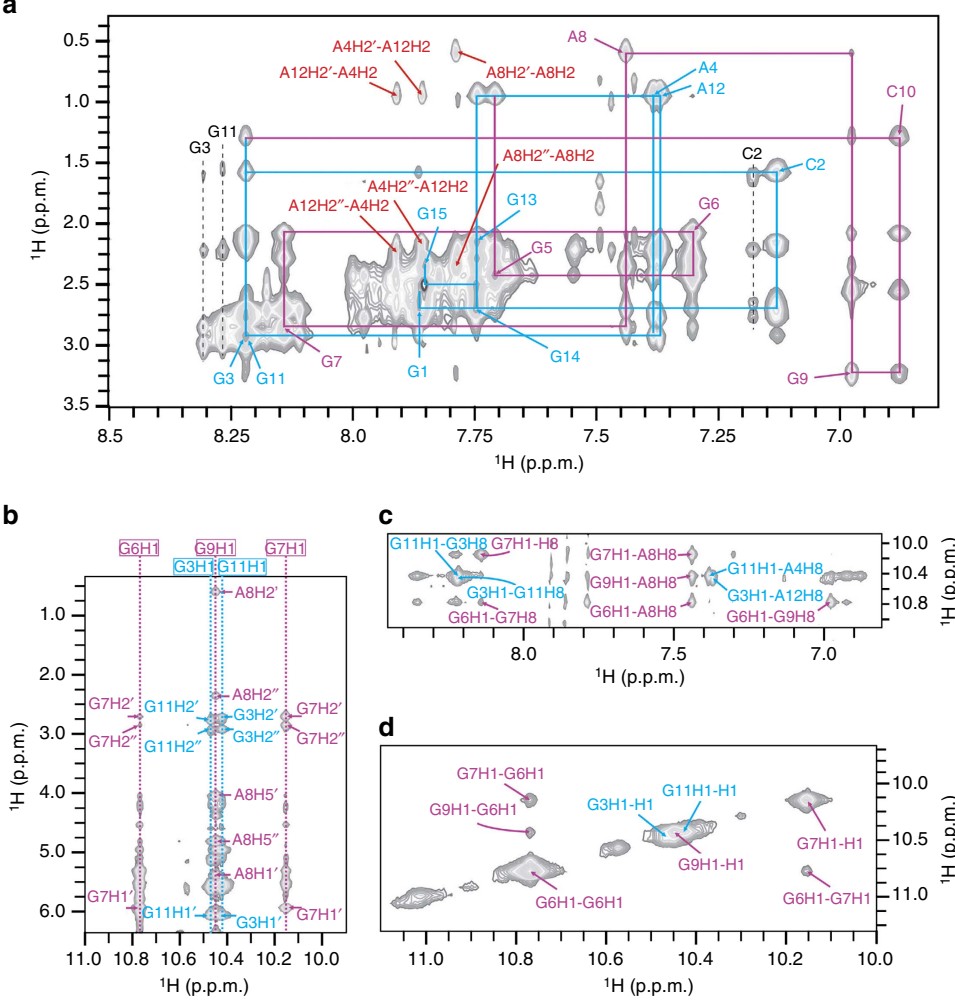

**Figure 4 | NOESY spectrum of the VK34 tetramer.** (**a**) H2′/H2″-aromatic region. The sequential walks from G1 to A4 and C10 to G15 are depicted in blue. The sequential walk from G5 to G9 is depicted in magenta. The NOE contacts between A4 and A12 residues as well as A8 H2′-A8 H2 and A8 H2″-A8 H2 NOE contacts are depicted in red. (**b**) Sugar-imino region. The dashed lines correspond to chemical shifts of imino proton resonances of G3, G6, G7, G9 and G11 residues. (**c**) imino-aromatic region. (**d**) imino-imino region. NOE contacts with fingerprints characteristic only for the VK34 tetramer are depicted in magenta. NOE contacts that have fingerprints similar to the VK34 dimer are depicted in blue. NOESY spectrum was recorded at 1.0 mM oligonucleotide concentration per strand, 150 mM NaCl, pH 6.0, 0 °C and at mixing time of 150 ms.

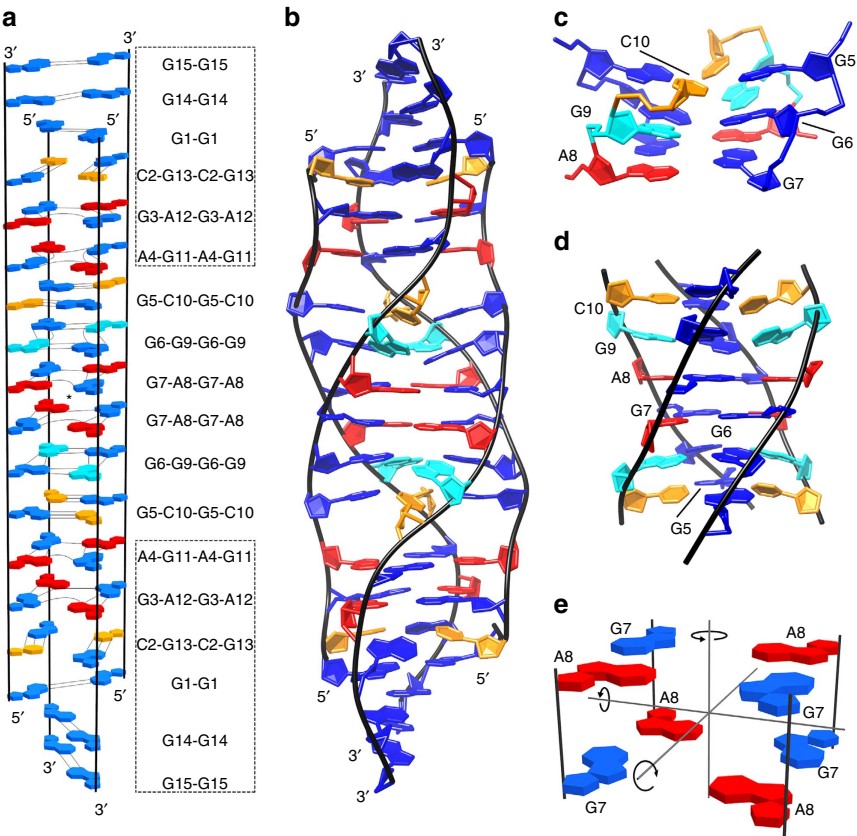

**Figure 5 | Schematic representation and the lowest-energy structure of the VK34 tetramer.** (**a**) Schematic representation of the VK34 tetramer. The asterisk marks the intersection point of three C$_2$ rotational axes of symmetry. (**b**) Cartoon representation of the overall lowest-energy structure (PDB ID: 5M2L). (**c**) G7-A8-G7-A8, G6-G9-G6-G9 and G5-C10-G5-C10 quartets. (**d**) View into a narrow groove of the central region. (**e**) Two stacked GAGA-quartets with C$_2$ rotational axes of symmetry. The guanine bases in *anti* orientation are coloured in blue, guanines in *syn* orientation in cyan, adenine in red and cytosine in orange. Hydrogen bonds are shown as dotted black lines. The dashed boxes in **a** highlight parts of the structure that adopt similar structural arrangements as in the dimeric structure.

axes (Fig. 5e). Such high degree of symmetry is rare and is not observed in G-quadruplexes, since stacked G-quartets possess only one C$_4$ axis of symmetry, nor in i-motifs that are composed of intrinsically asymmetric intercalated cytosine–cytosine$^+$ base pairs. The same C$_2$ axes that intersect two stacked G7-A8-G7-A8 quartets apply to the entire structure. A G6-G9-G6-G9 quartet is stacked on a G7-A8-G7-A8 quartet that is supported by the high number of NOE contacts between G6 and G7 as well as A8 and G9 residues (Fig. 5d and Supplementary Fig. 13). A high-intensity intraresidual NOE cross-peak between G9 H1' and H8 protons is in accordance with its *syn* orientation across the glycosidic bond. G6-G9-G6-G9 quartets therefore exhibit alternating *anti*(G6)–*syn*(G9) glycosidic bond orientations and are characterized by interchanging narrow and wide grooves. Additionally, NOE contacts between H1 and H8 protons of G9 and H1 protons of G6 are observed that are typical of G-quartets (Fig. 4c,d). Each G6-G9-G6-G9 quartet is sandwiched between G7-A8-G7-A8 and 'major groove' G5-C10-G5-C10 quartets (Figs 1d and 5c). In the latter, cytosine amino protons form hydrogen bonds with O6 guanine atoms of the neighbouring G-C base pair (Supplementary Fig. 14). The tetramer contains G5-C10-G5-C10 and C2-G13-C2-G13 major groove GCGC-quartets. The G5-C10 base pairs are formed between residues in different strands with respect to two C2-G13 base pairs and facilitate the transition from interchanging narrow and wide grooves in the central region to four medium grooves observed in the top and bottom parts of the tetramer. Even though G-C base pairs in the tetramer structure are predisposed for major groove GCGC-quartet formation,

protons of C2 (H6, H41), C10 (H41), G5 (H1) and G13 (H1) residues display two resolved resonances with minimal differences in chemical shifts, suggesting dynamic exchange between 'free' G-C base pairs and major groove GCGC-quartets. Comparison of the tetrameric and dimeric structures reveals that residues in the G1-A4 and G11-G15 segments adopt very similar structural arrangements in both folds (Figs 3a and 5b). These residues define the top and bottom parts of the tetramer characterized by A4-G11-A4-G11 and G3-A12-G3-A12 quartets that are continued by stacking of C2-G13-C2-G13 quartets. The G1-G1 and G14-G14 base pairs are arranged in a crisscross topology with the G15-G15 base pair stacked on the latter.

In the tetrameric VK34 structure the central channel created by GGGG-, GAGA- and GCGC-quartets is lined by carbonyl and amino groups that engage in hydrogen bonding interactions with each other, thus diminishing the repulsive electrostatic interactions usually associated with stacked G-quartets alone. Such a channel does not require cations for stabilization. In G-quadruplexes, at least two G-quartets are engaged in stacking interactions creating a central negatively charged channel lined by guanine carbonyl oxygen atoms. Monovalent cations such as K$^+$, Na$^+$ and NH$_4^+$ neutralize repulsive electrostatic interactions of the carbonyl oxygen atoms, thus critically stabilizing G-quadruplexes. Since Li$^+$ ions have high dehydration energies their coordination in the form of naked ions has never been observed in G-quadruplexes. The nature of the central channel in the VK34 tetramer makes it insensitive to the presence of cations. As G6-G9-G6-G9 quartets are the only two structural elements

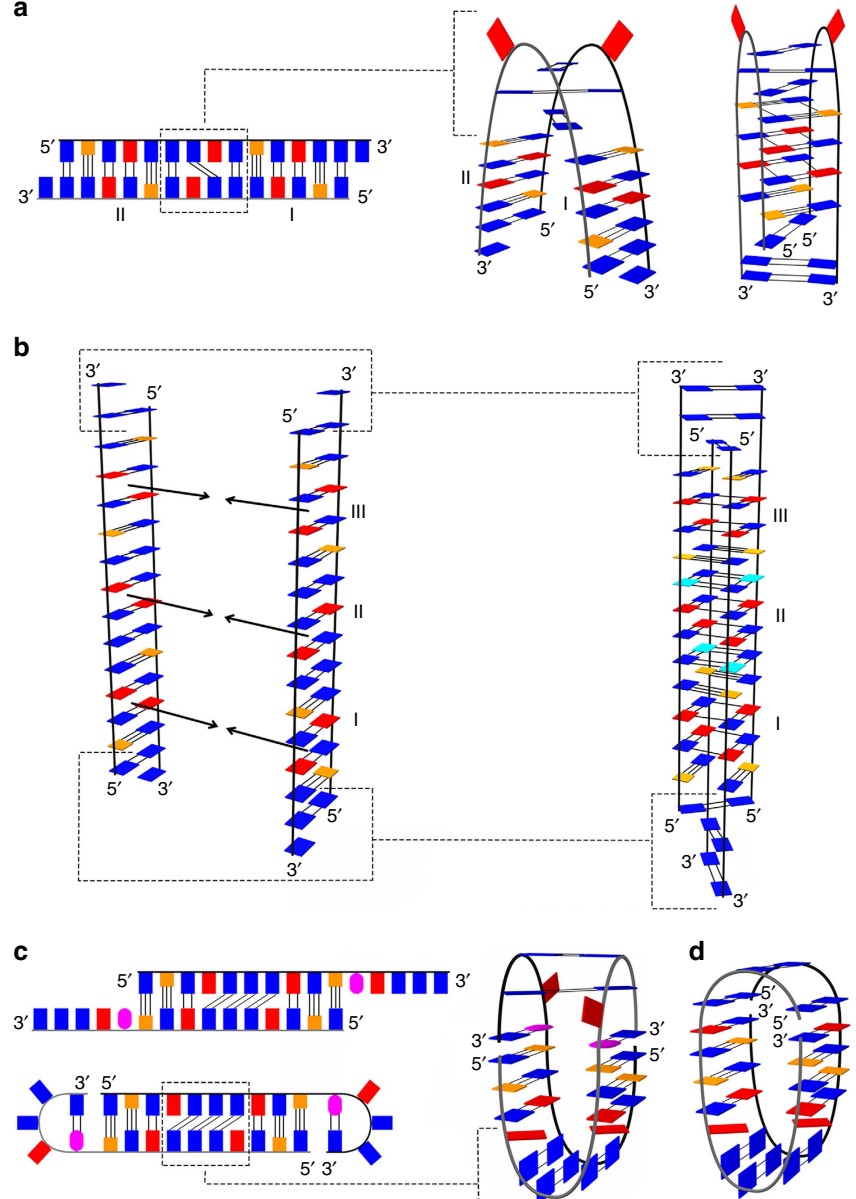

**Figure 6 | Simplified folding model of oligonucleotides with AGCGA repeats separated by GGG tracts.** (**a**) Folding model of the VK34 dimer. Roman numerals I and II indicate G-A base pairs that form the GAGA-core of the VK34 dimer. Completely folded VK34 dimer is shown on the right. (**b**) Folding model of the VK34 tetramer. The dashed lines show residues that form different base pairs in the VK34 tetramer than in the slipped duplex. Roman numerals I, II and III together with the arrows indicate G-A base pairs that associate into GAGA-quartets. (**c**) Folding model of VK34_I11. The dashed lines highlight different arrangements of two unpaired A4 residues separated by three G-G base pairs observed in a duplex compared with the VK34_I11 structure (right). (**d**) Topology of VK1. The guanine bases are shown in blue, guanines in *syn* conformation are indicated in cyan, inosine in magenta, adenine in red and cytosine in orange.

whose carbonyl oxygen atoms are not involved in hydrogen bonds with amino groups, they exhibit a higher degree of non-planarity, thus reducing their mutual repulsive interactions. It appears that although G6-G9-G6-G9 quartets do not require $K^+$, $Na^+$ and $NH_4^+$ ions for structural stabilization, their presence facilitates formation of the VK34 tetramer. Interestingly, molecular dynamics simulations under explicit water conditions showed that randomly placed $Na^+$ cations moved between G6-G9-G6-G9 and G7-A8-G7-A8 quartets inside the VK34 tetramer. $Na^+$ cations were retained closer to O6 atoms of the G6-G9-G6-G9 quartets than amino and O6 atoms of the G7-A8-G7-A8 quartets (Supplementary Fig. 15).

Oligonucleotides with adenine residues formally added at the 5'- (A_VK34), 3'- (VK34_A) and both ends (A_VK34_A) of the VK34 minimally affect the dimeric and tetrameric VK34 topologies as evidenced through comparison of respective 1D $^1$H and two-dimensional (2D) NOESY spectra (Supplementary Fig. 16).

**Unique structural motifs are reflected in helical parameters.** The family of 10 final structures of the VK34 dimer exhibits a rise between 3.4 and 3.8 Å, a typical value for right-handed DNA (Supplementary Fig. 17a,b). The two GAGA-quartets in the core

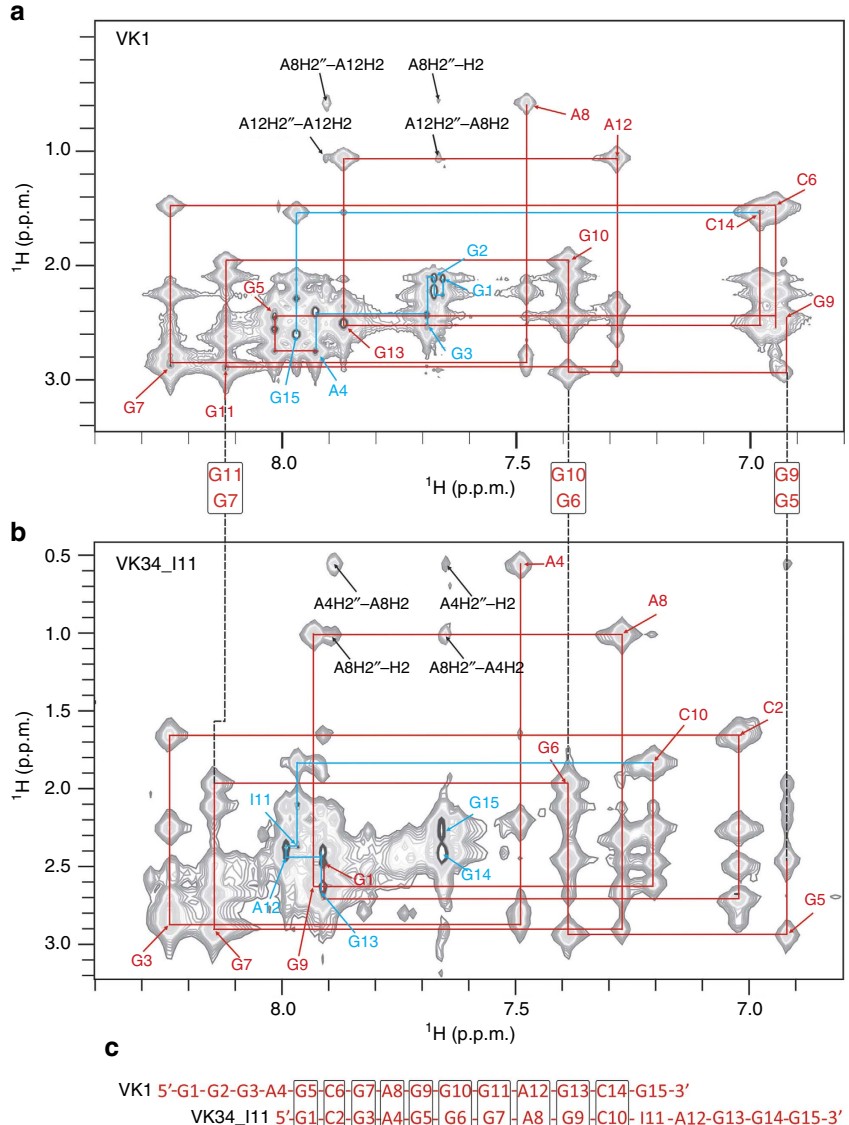

**Figure 7 | VK1 and VK34 folds have a very similar NOE fingerprint.** Comparison of H2′/H2″-aromatic regions of NOESY spectra of (**a**) VK1 and (**b**) VK34_I11. Cross-peaks with very similar NOE fingerprints in NOESY spectra of VK1 and VK34_I11 are indicated in red. NOE fingerprints without similarities are indicated in blue. (**c**) Sequences of VK1 and VK34_I11 oligonucleotides. Residues with similar NOE fingerprints are indicated by black frames. NOESY spectra (mixing time of 200 ms) of VK1 and VK34_11 folds were recorded at 1.0 mM oligonucleotide concentrations per strand and 100 mM LiCl, 0 °C and pH 6.0.

of the structure have a twist of 25° that is reduced to 5° at the G5-C10-G5-C10 quartet and the C2-G13 base pairs. G1-G1, G14-G14, G6-G6 and G9-G9 base pairs that are arranged in a crisscross topology are characterized by large twists between 50° and 60° (Supplementary Fig. 17b). Rise and twist are reflected in the circular dichroism (CD) spectrum of the VK34 dimer with a negative band at ∼245 nm and a positive band at ∼270 nm that are signatures of a right-handed stacked helix[17,18]. G1-G1 and G14-G14 as well as G6-G6 and G9-G9 base pairs arranged in a crisscross topology and characterized by large twists probably correspond to two additional shoulders at 280 and 290 nm (Supplementary Fig. 18). We observed slight buckling of G5-C10, G11-A4 and G15-G15 base pairs with values of 10°, 8° and 7°, respectively (Supplementary Fig. 17c). Higher buckling of 28° was observed for C2-G13 base pairs in addition to propeller twist of 14°. Analysis of α, β, γ, δ, ε, ζ and χ torsion angles of VK34 dimer showed distribution of backbone torsion angles in regions characteristic for A, B and Z DNA observed in crystal structures

of 96 DNA oligonucleotides rich in guanine, cytosine and adenine residues (Supplementary Fig. 17d)[19]. Such a broad distribution of α, β, ε and ζ angles is nicely correlated with a wide dispersion of [31]P NMR signals between 1.41 and −2.10 p.p.m. for the VK34 dimer (Supplementary Figs 19 and 21). The most downfield [31]P signals with chemical shifts of 1.41 and 1.39 p.p.m. were assigned to phosphate groups of A4 and A12 residues, respectively. The phosphate groups of their sequential neighbours G3 and G11 were assigned to upfield signals at −1.59 and −1.61 p.p.m., respectively. Such high [31]P chemical shift differences can be correlated with structural features of the two GAGA-quartets in a head-to-head orientation. Phosphate groups of G6 and G14 residues involved in crisscross topologies resonate at −0.24 and 0.32 p.p.m., respectively. Most residues in VK34 dimer exhibit South sugar conformation with the exception of C2, A4, G5, C10 and G13 that adopt North-type puckering (Supplementary Table 4). Their conformations correlate with surprisingly upfield [1]H NMR chemical shifts of C2 H2′, A4 H2′, G5 H2′, G5 H2″,

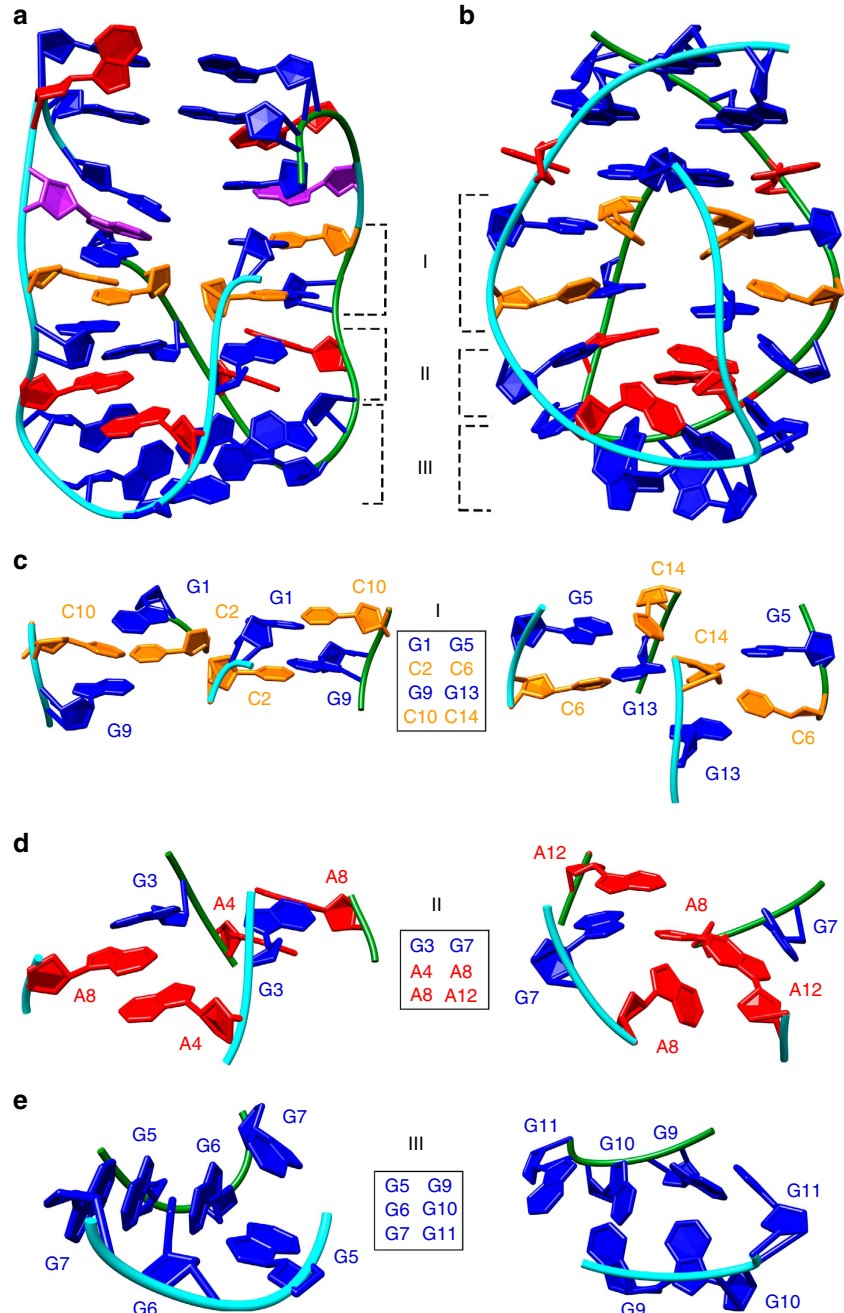

**Figure 8 | Comparison of VK34_I11 and VK1 structures.** (**a**) Overall representation of VK34_I11 structure. (**b**) VK1 structure. (**c**) Side views of G-C base pairs in VK34_I11 (left) and VK1 (right). (**d**) Insight into two G-A base pairs and the two A4 (left) and two A8 (right) residues orientated inside a hydrophobic pocket in VK34_I11 (left) and VK1 (right). (**e**) Side view of three G-G base pairs in fold-back arrangements in VK34_I11 (left) and VK1 (right). The roman numerals I, II and III refer to locations of the individual regions within the overall structure shown in **a**. The guanine residues are coloured blue, adenine red, cytosine orange and inosine purple. The two strands are coloured green and cyan.

C10 H2′ and A12 H2′ proton resonances (Supplementary Fig. 1). Structurally, this is probably due to tight packing and sugar-base stacking interactions (that is, C2 C2′-H2′/H2″···G3 purine, A4 C2′-H2′/H2″···G5 purine, A12 C2′-H2′/H2″···G13 purine) as well as close proximity of G5 and C10 ribose sugar moieties (Supplementary Fig. 5). The χ torsion angle values are typical for *anti* conformation for all residues with G3, A4 and G11 leaning to a *high anti* region (Supplementary Fig. 17d and Supplementary Table 4).

The VK34 tetramer exhibits a rise between 2.9 and 4.2 Å with the exception of A4-G11-A4-G11/G5-C10-G5-C10 steps with

values close to 5 Å (Supplementary Fig. 20a,b). The majority of base pair steps are characterized by twists between 10° and 27°. The G1-G1 and G14-G14 base pairs involved in crisscross topologies have twist values between 40° and 44°. Buckle values for G1-G1, G14-G14 and G15-G15 base pairs are between 13° and 28°, while propeller twist values are between 7° and 23° (Supplementary Fig. 20c). A buckle between 8° and 9° is observed for C2-G13 and G3-A12 base pairs. G5-C10 base pairs exhibit propeller twists of 11°. Most quartets are not planar and exhibit quartet buckling between 11° and 18° with the exception of the two C2-G13-C2-G13 quartets with values of ∼28°. Analysis of

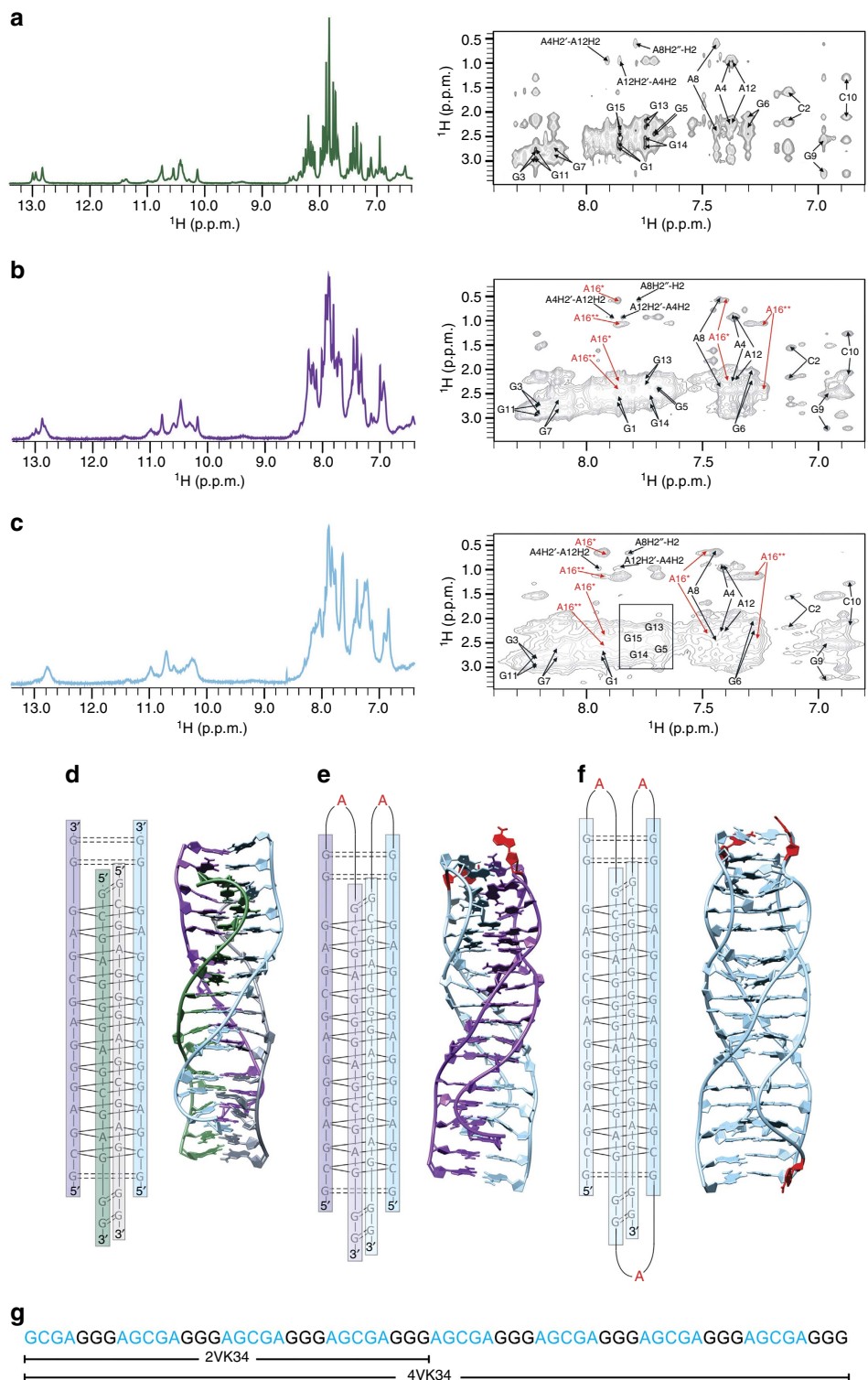

**Figure 9 | Folds very similar to VK34 tetramer are adopted also by 2VK34 and 4VK34 oligonucleotides.** Imino and aromatic regions of 1D $^1$H NMR and H2′/H2″-aromatic region of 2D NOESY spectrum for (**a**) VK34 tetramer, (**b**) 2VK34 fold and (**c**) 4VK34 fold. The 1D $^1$H and 2D NOESY spectra of 2VK34 and 4VK34 folds were recorded at 1.0 mM oligonucleotide and 100 mM LiCl concentrations, 0 °C and pH 6.0. The 1D $^1$H and 2D NOESY spectra of tetrameric VK34 fold were recorded at 1.2 mM oligonucleotide and 100 mM NaCl concentrations, 0 °C and pH 6.0. All spectra were recorded with 200 ms mixing times. (**d**) Tetrameric topology and high-resolution structure of VK34. (**e**) Dimeric topology and high-resolution structure of 2VK34. (**f**) Monomeric topology and high-resolution structure of 4VK34. (**g**) Oligonucleotide sequences of 2VK34 and 4VK34 oligonucleotides.

backbone torsion angles revealed a distribution similar to those found in the VK34 dimer (Supplementary Fig. 20d). G9 is found in *syn* while all other residues are in *anti* conformation. Sugar

moieties of the VK34 tetramer adopt South-type puckering with the exception of A8 that exhibits both North- and South-type conformations (Supplementary Table 5).

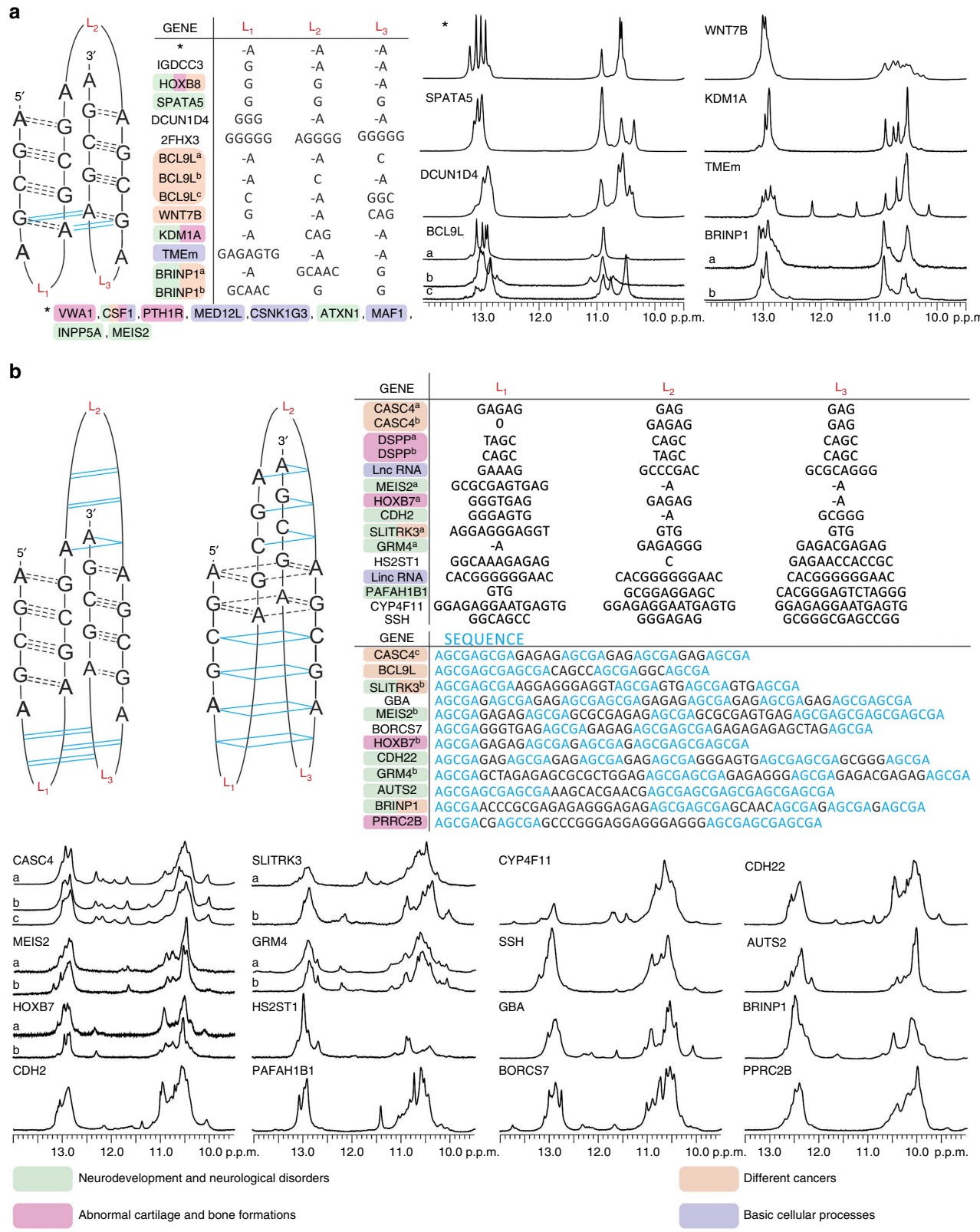

**Figure 10 | AGCGA repeats found in different human genes with NMR spectra and proposed topologies. (a)** List of genes and oligonucleotide sequences where 5′-AGCGA-3′ repeats are separated by few residues or experience adenine deletion between two sequential 5′-AGCGA-3′ repeats 5′-AGCG_AGCGA-3′ ( − A) with their proposed topology. Expansions of imino regions of 1D $^1$H NMR spectra that confirm formation of AGCGA-quadruplexes are shown. **(b)** Two proposed topologies adopted by oligonucleotides where four 5′-AGCGA-3′ repeats are separated by longer loops or that contain more than four 5′-AGCGA-3′ repeats. Respective imino regions of 1D $^1$H NMR spectra are shown. The spectra were recorded at oligonucleotide concentrations ranging from 0.6 to 1.1 mM, 100 mM LiCl concentration, pH 6.0 and 0 °C. Colour coding indicates implication of individual gene with family of diseases.

**Pre-folding states guide formation of AGCGA-quadruplexes.**
The VK34 dimer is formed under kinetically favoured conditions and spontaneously transforms into the tetrameric structure over time. This structural transition is accelerated in the presence of $K^+$, $Na^+$ or $NH_4^+$ cations or at increased oligonucleotide concentration. The tetrameric structure is thermodynamically favoured, since its transition back to the dimeric state can only be stimulated by large dilutions or by increasing the pH above 11 and temperatures above 50 °C. Melting curves showed temperatures of mid-transition ($T_{1/2}$) of ~43 and 42 °C for the VK34 dimer (100 mM LiCl) and tetramer (100 mM NaCl), respectively (Supplementary Fig. 22). No $T_{1/2}$ could be determined for the VK34 dimer at 4 µM oligonucleotide concentration per strand. Very broad melting with $T_{1/2}$ of ~45 °C was observed for the VK34 tetramer at 6 µM oligonucleotide concentration per strand. Hysteresis is more pronounced at lower oligonucleotide concentrations (Supplementary Fig. 22).

The transition between dimeric and tetrameric VK34 structures is possible since both structures share a common pre-folded state in form of a duplex, mostly stabilized by non-canonical base pairs, formed by two antiparallel VK34 oligonucleotides slipped by one residue (Fig. 6a,b). A pre-folded duplex can bend at two central A8 residues, causing the three G-G base pairs to arrange themselves in a crisscross topology, bringing the four G-A base pairs close enough to form two stacked GAGA-quartets. Additional G-G base pairs are formed at the 5′ and 3′ ends, while two G-C base pairs associate into a minor groove GCGC-quartet to form the dimeric VK34 structure. In contrast, association of two pre-folded duplexes leads to formation of a VK34 tetramer (Fig. 6b). Its stabilization originates from additional hydrogen bonds among G-A, G-G and G-C base pairs that in fact associate into GAGA-, GGGG- and GCGC-quartets. Additionally, three G-G base pairs at the 5′ and 3′ ends of the VK34 tetramer adopt a crisscross topology. Since pre-folded duplexes are pre-organized for formation of the tetrameric fold, its folding is fast.

With the intent to interrupt A4-G11-A4-G11 quartet formation and subsequently disturb the symmetry of the GAGA-core in VK34 structures, we substituted guanine at position 11 with an inosine residue (respective oligonucleotide is named VK34_I11, Fig. 6c). Unexpectedly, [1]H NMR spectrum of VK34_I11 was completely different from the spectra of dimeric and tetrameric VK34 folds. Since inosine differs from guanine only by absence of a C2-amino group, the effect is surprising. It has been suggested that inosine residues can influence DNA helicity that could interfere with formation of the pre-folded duplex[20]. Instead, great similarities were found between [1]H NMR spectra of VK34_I11 and VK1 (Supplementary Fig. 23)[17]. Similar to the VK1 structure, VK34_I11 folds into a dimer that has also been confirmed with the use of analytical ultracentrifugation (Supplementary Fig. 3). The G1-C10 segment of VK34_I11 exhibits an almost identical NOESY fingerprint as the G5-C14 segment of VK1 (Fig. 7). Even though the similarities in the NOESY spectra enabled almost complete assignment of 2D NOESY spectra of VK34_I11, unambiguous assignment of its H1 proton resonances was confirmed with the help of [15]N-HSQC spectra acquired on partially (10%) residue-specifically [15]N-labelled oligonucleotides (Supplementary Fig. 24). VK1 and VK34_I11 oligonucleotides fold into well-defined symmetric dimeric structures, with four G-C base pairs, two G-A base pairs, three G-G base pairs and unpaired adenine residues located in almost identical orientations (Figs 6c,d and 8, PDB ID: 5M4W and Supplementary Table 3). The remaining residues adopt different structural arrangements such as an intramolecular G-I and a couple of G-G base pairs with two adenine residues forming loop regions in VK34_I11 compared with three G-G and two G-A base pairs observed in VK1.

The 5′-d(GCGAGGGAGCIAGGG)-3′, VK34_I11, and 5′-d(GGGAGCGAGGGAGCG)-3′, VK1, sequences can form similar structures since two VK34_I11 oligonucleotides align to form a duplex that encompasses a segment with the same composition and distribution of base pairs as found in the VK1 structure (Fig. 6c,d). The duplex is slipped by five residues with respect to the completely base-paired topology and contains 3′ overhangs composed of an inosine, adenine and three guanine residues. The overhangs form 3′ loops comprising one adenine and two guanine residues by establishing intramolecular Hoogsteen G-I base pairs. A slipped duplex bends in a way that three G-G base pairs arrange themselves around two unpaired adenine residues in a fold-back topology and two 3′ loops are stabilized by a couple of G-G base pairs.

**Multiple AGCGA-repeat constructs adopt unimolecular folds.**
We expected that unimolecular pre-folded states could also be formed by longer oligonucleotides, and hence we prepared 2VK34 and 4VK34 constructs in which two and four 15-nucleotide sequences of VK34 are connected by single adenine residues, respectively. The 1D [1]H and 2D NOESY spectra of the 2VK34 and 4VK34 constructs display almost identical fingerprints as the tetrameric fold of VK34 (Fig. 9a–c). Analysis of NOESY spectra showed that the linking adenine residues in the 2VK34 and 4VK34 constructs enable bimolecular and unimolecular folds by formally connecting the 5′ and 3′ ends of individual molecules of VK34 (Fig. 9d–f). Bands with similar mobilities for the VK34 tetramer, 2VK34 dimer and 4VK34 monomer were detected in a native PAGE gel assay (Supplementary Fig. 7). Molecularity of 2VK34 and 4VK34 structures was confirmed by analytical ultracentrifugation (Supplementary Fig. 3). The discrepancy in sedimentation coefficients observed for the VK34 tetramer in the presence of $Na^+$ cations compared with the 2VK34 dimer and 4VK34 monomer can be attributed to considerable difference in concentrations required to stabilize the tetrameric form, or to different hydrodynamic properties due to high flexibility of the 5′- and 3′-ends of the VK34 tetramer.

It is important to note that sequences of 2VK34 and 4VK34 oligonucleotides comprise alternating 5′-AGCGA-3′ and 5′-GGG-3′ repeats and could be expected to form G-quadruplexes whose formation is not observed (Fig. 9g).

**An overlooked structural motif.** Analysis of the VK34 dimer and tetramer, 2VK34, 4VK34, VK1 and VK34_I11 structures reveals that the basic requirement for formation of AGCGA-quadruplexes is the presence of four 5′-AGCGA-3′ repeats located in one or more strands. According to the structure of the AGCGA-core we can separate AGCGA-quadruplexes into two distinct types. In the first type, GAGA- and GCGC-quartets are stacked and represent the core of tetrahelical structure. In the second, four G-C base pairs are located in the centre of the structure and G-A base pairs are stacked on G-G or G-A base pairs. We checked for occurrence of oligonucleotides that could fold into unimolecular AGCGA-quadruplexes in DNA by performing a human genome-wide search for sequences that correspond to the 5′-AGCGA($N_{1-20}$)AGCGA($N_{1-20}$)AGCGA$N_{1-20}$)-AGCGA-3′ motif. The search revealed 146 sequences with 41 (in addition to VK1, VK2, VK34, 2VK34 and 4VK34) of them found in promoter regions and transcription start sites of known genes as well as CTCF (CCCTC-binding factor) binding sites and copy number variation regions. Fourteen of the identified genes including *ATXN1*, *MEIS2*, *KDM1A*, *GRM4*, *PAFAH1B1*, *CDH22* and *AUTS2* are linked to neurodevelopment and neurological disorders such as autism, obsessive compulsive disorder, epilepsy,

schizophrenia, mental retardation, KBG syndrome and Miller–Dieker syndrome (Fig. 10)[21–34]. Seven genes including *VWA1*, *PTH1R*, *KDM1A* and *CSF1* have been related to abnormal cartilage and bone formations, with the *CSF1* gene also linked to tenosynovial giant cell tumour[35–39]. Interestingly, clinical situations have shown that patients with neurologic disorders exhibit localized bone changes and altered fracture healing with excessive callus formation[40]. It is likely that many of the genes involved in neurologic disorders and abnormalities in bone and cartilage development affect similar biochemical processes. A prime example is the *CDH2* gene that is involved in development of the nervous system, formation of cartilage and bone and is additionally connected to obsessive–compulsive, Tourette and related disorders[32]. Eight genes including *BCL9L*, *CASC4*, *HOXB8* and *BRINP1* have been linked to different types of cancer[41–45]. Six genes are associated with basic cellular processes such as regulation of RNA polymerases and intracellular transport[46–51].

We recorded 1D $^1$H NMR and CD spectra of 41 oligonucleotides from the above-mentioned biologically relevant regions in the presence of only Li$^+$ ions (Fig. 10 and Supplementary Figs 25–27). 1D $^1$H NMR spectra of all samples show a fingerprint in the H1 proton region that is similar to VK1, VK34, VK34_I11, 2VK34 and 4VK34 folds (Fig. 10). CD spectra with a negative band at ~245 nm and a positive band at ~270 nm as well as additional bands between 271 and 310 nm are easily distinguishable from the spectra of other higher-order DNA structures (Supplementary Figs 26–28).

In accordance with assignment of the two shoulders at 280 and 290 nm in CD spectra of VK1 to fold-back loops stabilized by G-G base pairs, bands at wavelengths higher than 270 nm correspond to different G-G and G-A base pair arrangements adopted by different folds of AGCGA-rich sequences.

Oligonucleotides where four 5′-AGCGA-3′ repeats are separated by a few or no residues adopt AGCGA-quadruplexes, where four G-C base pairs are located in the centre of the structure and G-A base pairs are stacked on G-G or G-A base pairs that are not part of the AGCGA-core (Fig. 10a). Interestingly, such topology also applies to the simplest sequence containing four 5′-AGCGA-3′ repeats, 5′-d(AGCGAGCGAGCGAGCGA)-3′, located in regulatory regions of nine genes that shows well-resolved H1 proton resonances. In this case, adenines that formally belong to edgewise loops are also involved in G-A base pairs. Oligonucleotides where four 5′-AGCGA-3′ repeats are separated by longer segments mostly composed of guanine and adenine residues or that contain more than four 5′-AGCGA-3′ repeats can adopt more complex AGCGA-quadruplex topologies discussed above (Fig. 10b).

We report results of structural studies on AGCGA-quadruplexes, a new family of tetrahelical structures that are characterized by arrangements of four 5′-AGCGA-3′ repeats from one or more oligonucleotides. Using NMR-derived structural data we demonstrate that AGCGA-quadruplexes are stabilized by G-A and G-C base pairs forming GAGA- and GCGC-quartets, respectively. Alternatively, the core of the AGCGA-quadruplex consists of four central G-C base pairs with continued stacking of G-A base pairs and GGG tracts. Residues in the tetrahelical core of the structure with antiparallel topology are connected with edge-type loops stabilized mostly by G-G base pairs in N1-carbonyl symmetric geometry. Sequences of alternating 5′-AGCGA-3′ and 5′-GGG-3′ repeats could be expected to form G-quadruplexes but form AGCGA-quadruplexes instead. Formation of AGCGA-quadruplexes is critically affected and expedited by specific pre-folded structures. 5′-AGCGA-3′-repeat sequences and their respective structural family are potentially of great biological relevance since they are adopted by 46

oligonucleotides found in regulatory regions of 38 different human genes connected to neurodevelopment and neurological disorders, abnormal cartilage and bone formations, cancer and regulation of basic cellular processes. Unique structural features of AGCGA-quadruplexes together with their lower sensitivity to cations and pH variation support the biological relevance of structures formed by 5′-AGCGA-3′-repeat sequences in regulatory regions of genes responsible for basic cellular processes and in neurological disorders, cancer and abnormalities in bone and cartilage development.

## Methods

**Sample preparation.** The isotopically unlabelled, residue-specific low-enrichment (10% $^{15}$N-labelled and 10% $^{13}$C, $^{15}$N-labelled) and residue-specific completely (100%) D8-labelled oligonucleotides were synthesized on K&A Laborgeraete GbR DNA/RNA Synthesizer H-8. In all cases, standard phosphoramidite chemistry was used. The synthesis was done with Link technologies iBu-dG, iBu-dA and universal Q SynBase columns. Deprotection was done with the use of aqueous ammonia for 12 h at 55 °C when using iBu-dG and iBu-dA and at 80 °C when using universal Q SynBase columns. Residue-specific completely (100%) D8-labelled VK34 oligonucleotides were deprotected with ND$_4$OD at 55 °C for 12 h. The VK34 oligonucleotide that had all guanine and adenine residues completely (100%) D8-labelled was prepared by exchanging the guanine and adenine H8 atoms with deuterium atoms in ND$_4$OD at 80 °C for 12 h. Samples were purified and desalted with the use of Millipore Stirred Ultrafiltration Cell model 8010 and Amicon Ultra-15 Centrifugal Filter Devices to give NMR samples with concentrations between 0.4 and 2.4 mM per strand. The samples were prepared in the presence of LiCl with varying concentrations from 10 to 100 mM as well as 100 mM concentrations of NaCl, KCl and NH$_4$Cl. pH value was set between 6.0 and 6.5 with the use of LiOH and HCl. Extra care was taken to ensure that only the specified cations were present in NMR samples.

**NMR spectroscopic experiments.** All NMR experiments were performed on Agilent-Varian NMR Systems 600 and 800 MHz spectrometers equipped with triple-resonance HCN cryogenic probes in the temperature range from 0 to 25 °C. The vast majority of spectra were recorded at 0 °C on samples in 90% H$_2$O and 10% D$_2$O. 1D $^{15}$N-edited HSQC as well 2D $^{15}$N- and $^{13}$C-edited heteronuclear multiple-quantum correlation spectroscopy experiments were performed on 10% residue-specific $^{15}$N-, $^{13}$C-labelled oligonucleotides. DPFGSE_NOESY with mixing times of 40, 80, 100, 200 and 250 ms were acquired on unlabelled and D8-labelled oligonucleotides. $^1$H–$^{31}$P COSY and 1D $^{31}$P spectra were acquired in 100% D$_2$O. Diffusion experiments were performed using 30 different gradient strengths (0.49–29.06 G cm$^{-1}$). NMR spectra were processed and analysed using VNMRJ (Agilent) and Sparky (UCSF) software.

**CD spectroscopy.** CD experiments were carried out on an Applied Photophysics Chirascan CD spectrometer over 190–330 nm and 200–320 nm wavelength ranges. All measurements were made in 0.01 cm path-length quartz cells. The oligonucleotide concentrations were between 0.5 and 1.0 mM.

**Ultraviolet spectroscopy.** The oligonucleotide concentration was determined by measuring the ultraviolet absorbance at 260 nm on a Varian Cary 100 Bio with 1.0 cm path-length cells. The extinction coefficient of VK34 is 155,200 M$^{-1}$ cm$^{-1}$.

**Native PAGE.** The 10% polyacrylamide gels (acrylamide/bis-acrylamide 19:1) containing TBE (Tris/Borate/EDTA buffer) supplemented with 100 mM LiCl or 200 mM NaCl were prepared. Thermo Scientific GeneRuler Ultra Low Range DNA Ladder (300–10 bp) designed for 10% polyacrylamide gels was loaded on each gel. Oligonucleotide concentrations were between 40 and 100 μm. We ran the gels with 4 W and 100 V at 4 °C.

**Restraints and structure calculations.** Distance restraints (force constant 20 kcal mol$^{-1}$ Å$^{-2}$) used in structural calculations were obtained from NOESY spectra recorded at 200 ms (VK34 dimer, 2VK34, 4VK34 and VK34_I11) and 250 ms (VK34 tetramer) mixing time in 90% H$_2$O and 10% D$_2$O. The H2′/H2″ cross-peaks could not be used as references because we revealed that they experience a different NOE regime than cross-peaks in the rest of the spectrum. The cytosine H5-H6 cross-peak severely overlapped with the water signal and also could not be used as a reference. We had to rely on averaging the volumes of intra-nucleotide H8-H1′ NOE correlations of the residues that clearly exhibited an *anti* orientation and referencing the averaged volumes to a value of 3.9 Å. With the help of this reference, we classified the remaining signals as strong (1.8–3.6 Å), medium (2.6–5.0 Å), weak (3.5–6.5 Å) and very weak (4.5–7.5 Å). Torsion angle restraints (force constant 200 kcal mol$^{-1}$ rad$^{-2}$) along glycosidic bonds (torsion angle χ) for residues in *anti* orientations were set between 170° and 280° for purines and 170°

and 310° for pyrimidines. Planarity restraints (force constant 50 kcal mol$^{-1}$ rad$^{-2}$) for GAGA-quartets and G-G base pairs in N1-carbonyl symmetric geometry were used. Structure calculations were performed using AMBER 14 program suites and parmbsc0 version of the force field with the 2012 parmχOL4 and 2013 parmε/ζOL1 modification[52–55]. The initial extended single-stranded DNA structure was obtained using the leap module of AMBER 14. A total of 100 structures were calculated in 100 ps of NMR restrained simulated annealing simulations using the generalized Born implicit model. The SHAKE algorithm with a tolerance of 0.00005 Å for hydrogen atoms was used. The cutoff for nonbonded interactions was 8 Å. Ten lowest-energy structures calculated with simulated annealing were subjected to a maximum of 100,000 steps of steepest descent energy minimization (Supplementary Figs 29–31). Figures were visualized and prepared with UCSF Chimera software[56].

**Molecular dynamics refinement in explicit solvent.** The 10 best structures of the VK34 dimer, VK34 tetramer and VK34_I11 were refined further in explicit water solvent. Each structure was placed in a cuboid box of TIP3P water molecules with the box border at least 10 Å away from any atoms of the DNA. Extra Li$^+$ or Na$^+$ ions were added to neutralize the negative charges of DNA. After the models were built, the cation positions were randomized using CPPTRAJ by swapping random water and ion positions in a way that no cation was closer than 4 Å to another and all cations were further than 6 Å away from DNA to avoid any bias created by the initial placement of the ions[57]. The simulations were performed with the CUDA version of pmemd module of AMBER 14. The system was first minimized with harmonic potential position restraints (25 kcal mol$^{-1}$ Å$^{-2}$) used on DNA with over 500 steps of steepest descent minimization followed by 500 steps of conjugated gradient minimization. The system was then heated from 100 to 300 K over 50 ps under a constant volume while maintaining 25 kcal mol$^{-1}$ Å$^{-2}$ position restraints on DNA. Next, the system was equilibrated for 100 ps with 5 kcal mol$^{-1}$ Å$^{-2}$ position restraints on DNA and 200 ps with NMR restraints (without planarity restraints) at 300 K and 1 atm. Pressure coupling used during equilibration was set to 0.2. The production simulation with NMR restraints was carried out at constant pressure of 1 atm and constant temperature of 300 K maintained using Langevin dynamics with a collision frequency of 2.0. Periodic boundary conditions were used and electrostatic interactions were calculated by the particle mesh Ewald method with the nonbonded cutoff set to 9 Å (ref. 58). The SHAKE algorithm was applied to bonds involving hydrogens, and a 1 fs integration step was used[59,60]. The production run was carried out for continuous 5 ns and snapshots were written at every 1 ps. Finally, the system was further minimized for 5,000 steps. Structural refinement under explicit solvation conditions showed that structures of the VK34 dimer and tetramer as well as VK34_I11 were not changed compared with the implicit solvation model (root mean square deviation <2 Å, Supplementary Fig. 32 and Supplementary Tables 6–8). Additionally, we evaluated impact of base pair planarity restraints and concluded that they did not affect geometries in structural ensembles of the VK34 dimer, VK34 tetramer and VK34_I11 in explicit water conditions that included randomly placed Li$^+$ or Na$^+$ ions.

**Data availability.** The coordinates of dimeric (PDB ID: 5M1L) and tetrameric (PDB ID: 5M2L) structures adopted by VK34, and VK34_I11 (PDB ID: 5M4W) were deposited to the protein data bank. All data generated and analysed during this study are included in this published article (and its Supplementary Information files) or available from the authors on reasonable request.

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

## Acknowledgements

This work was supported by the Slovenian Research Agency (Grants P1-0242 and J1-6733). We gratefully acknowledge Professors Jonathan B. Chaires and William L. Dean, James Graham Brown Cancer Center, University of Louisville, Louisville, USA, for their help in performing AUC studies.

## Author contributions

V.K. and J.P. designed the experiments, interpreted data and wrote the manuscript. V.K. carried out the experiments.

## Additional information

**Competing interests:** The authors declare no competing financial interests.

**Publisher's note**: 

