## [Peer Review File · Nature Communications]

Reviewers' Comments:

Reviewer #1 (Remarks to the Author)

This is a highly significant manuscript with a substantial body of new data reported. The authors are reporting new structural motifs for tetrahelical nucleic acid structures from an NMR perspective. The characterization of these new folds is limited to mostly structural work and further biophysical analysis would have enhanced the results. However, the structures that they report are in themselves significant. As it is the structures they are reporting and commenting on, the relevant structure files should have been available for review. Overall, this is an exciting addition to the quadruplex/tetrahelical field.

Some comments and things that need to be addressed follow.

The statement "Unique structural features of AGCGA-quadruplexes together with their lower sensitivity to cation and pH variation support biological relevance of structures" is without significant justification and should be modified/deleted as the biological "relevance" of these structures is completely unknown at this stage.

In several places the English is a little disjointed due to missing articles or prepositions. Also, some typos are present "staling" is stalling. Correct referencing is needed Page2:line50 "One of the proposals predicts", where is this from?

The original biological relevance to PLEKHG3 is now lower as it is a AGCGA repeat reported not a GGGAGCG repeat. However, as stated, the possible AGCGA repeat sequences are present in the genome.

The 1H spectrum of VK34 (Sup. Fig 1) is extremely clean and symmetrical. A remarkable piece of data!

It would have been better to validate the dimeric nature of VK34 using analytical ultracentrifugation rather than gel analysis as this has been shown to be potentially unreliable or misleading. This is also true for the subsequent higher order structures reported.

The PDB files 5M1L and 5M2L are on hold so the reviewers cannot examine the structural aspects that are being reported. For manuscripts that are reporting new structures, it should be required to supply the PDB files for review.

It's stated that that all residues (bases) "assume anti orientations across glycosidic bonds" and the G3-A12-G3-A12 and G11-A4-G11-A4 quartets are in a "head to head" orientation. The full glycosidic torsion angle NMR data and subsequent structural angle analysis should be reported in Supplementary Data and commented on for all bases as there may be some unusual space explored. As it is, from Fig. 2., it is difficult/impossible to visualize that the glycosidic angles are all anti, in fact they look otherwise. As there were glycosidic angle restraints in place (170-310 deg), did this artificially distort the sugar and backbone? Following this, it would be beneficial to report the full analysis of the structure for sugar and standard base, base-pair, base steps, and strand parameter analysis in Supplementary Data and comment on it as these are new kinds of tetrahelical structures.

Page8-9 It is stated "The tetrameric structure is thermodynamically favored and its transition back to the dimeric state can be stimulated by increasing pH above 11 and temperatures above 50 °C." Further characterization of the dimer and tetramer should be included. What are the relative

melting temperature for both to put this statement in context.

Figure 5a is incorrect with respect to the placement of arrows for the monomer strand in the duplex to the folded tetrahelical form.

Methods: The authors use AMBER14 for the structural characterization. What is the actual experimental justification for the planarity restraints? It would have been interesting to refine the structures in explicit solvation conditions with the eventual reduction and elimination of restraints.

Reviewer #2 (Remarks to the Author)

In this study, Kocman and Plavec present NMR structures of a family of novel four-stranded nucleic acid structures formed in sequences with GCGAGGG repeats, which appear to be stabilized by GAGA- and GCGC-quartets. GAGA- and GCGC- quartets are distinct from the previously characterized Watson-Crick base pairs found in B-DNA or Hoogsteen G-quartets observed in G-quadruplexes. Their detailed characterization reveals many unique and surprising features which could not be predicted otherwise. The presented NMR data is extensive and the NMR structure determination is thorough in detail and of high quality. The approach taken is rational and straightforward. Notably, the authors determined the high resolution structures for multiple experimental DNA systems and provided additional support these with mutational and CD data. These structures are distinctly different from the four-stranded DNA structures they previously reported in closely related sequences of GGGAGCG repeats, which are stabilized exclusively by non-canonical base pairing. The demonstration of GAGA- and GCGC-quartets as a stable structure motif broadens the known space in which non-canonical nucleic acid structures may be formed. Significantly, the authors found that sequences capable of forming such four-stranded DNA structures, which they named AGCGA-quadruplexes, exist in the regulatory regions of 38 human genes connected to multiple diseases, suggesting that such DNA structures may carry implications in disease. The novel and unique structure of these tetrahelical structures adds valuable contributions both to the literature-base of structural biology and to the general understanding of nucleic acid structures, and in my opinion, is appropriate for publication in Nature Communications.

I have some comments for the authors to consider.

1. All the sequences in this start and end with guanine. Have the authors tested the effect of flanking DNA?
2. As this structure contains a number of novel base-pairing schemes, I think it would be helpful to have schematic representations of of all observed quartets and base-pairs as chemical structures including their hydrogen bonding.
3. On page 9, line 269-74, the description of the folding model for the formation of VK34 tetramer was unclear. In line 272, should "Fig. 4a and 4b" be "4a and 4d"?
4. On page 11, in the first paragraph, for the longer oligonucleotides 2VK34 and 4VK34, it would be helpful to include the 1D proton NMR spectra in addition to the 2D data shown in supplementary Fig. 18.
5. In supplementary Fig. 7, several imino protons are shown as doublets or multiplets. A discussion would be helpful.
6. Figure 6 is very intriguing and significant. The proposed intramolecular AGCGA-quadruplexes are mainly built upon the two core GCGC-quartets sandwiched by GAGA-quartets (shown in 6a and 6b left), which has been found in different human genes. This is different from the core of both

VK34 dimer and tetramer. It would be very nice to see the molecular structure, or at least confirmation of the folding structure, of such an intramolecular AGCGA-quadruplex.

Revision of manuscript NCOMMS-16-25550-T

We thank both reviewers for positive comments and constructive suggestions that helped us in revising our manuscript. Please find below our responses to all comments from both reviewers.

Reviewer #1

This is a highly significant manuscript with a substantial body of new data reported. The authors are reporting new structural motifs for tetrahelical nucleic acid structures from an NMR perspective. The characterization of these new folds is limited to mostly structural work and further biophysical analysis would have enhanced the results. However, the structures that they report are in themselves significant. As it is the structures they are reporting and commenting on, the relevant structure files should have been available for review. Overall, this is an exciting addition to the quadruplex/tetrahelical field.

Some comments and things that need to be addressed follow.

The statement “Unique structural features of AGCGA-quadruplexes together with their lower sensitivity to cation and pH variation support biological relevance of structures” is without significant justification and should be modified/deleted as the biological “relevance” of these structures is completely unknown at this stage.

The statement on potential relevance of AGCGA-quadruplexes has been modified on p. 1 of the revised manuscript. However, there are many reports in the literature (refs. 21-51 in the revised manuscript) where several genes implicated in diseases listed in Fig. 10 have AGCGA repeats located in their regulatory regions.

In several places the English is a little disjointed due to missing articles or prepositions. Also, some typos are present “staling” is stalling. Correct referencing is needed Page2:line50 “One of the proposals predicts”, where is this from?

We thank anonymous reviewer for his help in improving grammar and thus making our manuscript easier to follow. The typos were corrected and the sentence line 50 on p. 2 of the original manuscript has been rephrased in the revised manuscript.

The original biological relevance to PLEKHG3 is now lower as it is an AGCGA repeat reported not a GGGAGCG repeat. However, as stated, the possible AGCGA repeat sequences are present in the genome.

The original relevance of our earlier report on tetrahelical structures adopted by VK1 and VK2 oligonucleotides is not reduced. In fact, VK1 and VK2 structures are members of the AGCGA-quadruplex family, which is described on p. 18 of the revised manuscript.

The ¹H spectrum of VK34 (Sup. Fig 1) is extremely clean and symmetrical. A remarkable piece of data!

We are grateful for a nice comment.

It would have been better to validate the dimeric nature of VK34 using analytical ultracentrifugation rather than gel analysis as this has been shown to be potentially unreliable or misleading. This is also true for the subsequent higher order structures reported.

Molecularity of dimeric and tetrameric structures adopted by VK34 as well as 2VK34, 4VK34 and VK34_I11 was confirmed with the use of analytical ultracentrifugation. The results are presented in Supplementary Figure 3 and discussed on pages 3, 6, 13 and 16 in the revised manuscript. Noteworthy, evaluation of molecularity by sedimentation values obtained by analytical ultracentrifugation are in perfect agreement with gel analysis and translational diffusion coefficients determined by NMR.

The PDB files 5M1L and 5M2L are on hold so the reviewers cannot examine the structural aspects that are being reported. For manuscripts that are reporting new structures, it should be required to supply the PDB files for review.

According to the accepted practices structures were deposited in PDB and will be released to the public immediately after manuscript is accepted for publication.

Please allow us to express a humble opinion. After consultation with some eminences in the field of structural biology (speakers at a NMR conference in Paris, January 2017) it appears that community does not feel comfortable to share structural coordinates before publication is guaranteed.

It's stated that that all residues (bases) "assume anti orientations across glycosidic bonds" and the G3-A12-G3-A12 and G11-A4-G11-A4 quartets are in a "head to head" orientation. The full glycosidic torsion angle NMR data and subsequent structural angle analysis should be reported in Supplementary Data and commented on for all bases as there may be some unusual space explored. As it is, from Fig. 2., it is difficult/impossible to visualize that the glycosidic angles are all anti, in fact they look otherwise. As there were glycosidic angle restraints in place (170-310 deg), did this artificially distort the sugar and backbone? Following this, it would be beneficial to report the full analysis of the structure for sugar and standard base, base-pair, base steps, and strand parameter analysis in Supplementary Data and comment on it as these are new kinds of tetrahelical structures.

We now report the full glycosidic torsion angle and subsequent structural angle analysis as well as analysis of the sugar puckering and standard base-pair and base steps parameters in Supplementary Figures 17 and 20, and Supplementary Tables 4 and 5. These parameters are discussed in the special section on p. 10-11 of the revised manuscript. In short, perusal of the above figures and tables demonstrates that all residues adopt anti conformation along glycosidic bonds with G3, A4, G11 and A12 residues that form the two GAGA-quartets adopting a high anti conformation.

Torsion angle restraints (force constant $200 \text{ kcal mol}^{-1} \text{ rad}^{-2}$) along glycosidic bonds for residues in *anti* orientation were set between 170° and 280° for purines and between 170°

and 310° for pyrimidines. This is clearly stated in the methods section on p. 25 of the revised manuscript. We did not observe any artificial distortions of sugar and backbone.

Page8-9 It is stated “The tetrameric structure is thermodynamically favored and its transition back to the dimeric state can be stimulated by increasing pH above 11 and temperatures above 50 °C.” Further characterization of the dimer and tetramer should be included. What are the relative melting temperatures for both to put this statement in context?

Further characterization of dimeric and tetrameric folds of VK34 that form in the presence of 100 mM LiCl and NaCl, respectively utilizing concentration-dependent UV melting experiments is presented in Supplementary Figure 22 and discussed on p. 11 of the revised manuscript.

Figure 5a is incorrect with respect to the placement of arrows for the monomer strand in the duplex to the folded tetrahelical form.

In order to avoid possible confusion, arrows were replaced with roman numerals to more clearly show the corresponding G-A base pair regions in the extended and fold-back topologies. Additionally, illustrations presented in the previous Figure 5 have been improved and are now located in separate Figures 6 and 9 of the revised manuscript.

Methods: The authors use AMBER14 for the structural characterization. What is the actual experimental justification for the planarity restraints? It would have been interesting to refine the structures in explicit solvation conditions with the eventual reduction and elimination of restraints.

Potential impact of planarity restraints on the calculated structures has been (re)evaluated. We have ran 5 ns MD simulations of all high-resolution structures without planarity restraints under explicit water conditions that included Li⁺ and Na⁺ ions. Final structures showed only minimal differences (Supplementary Tables 6, 7 and 8) compared to the structures calculated in implicit solvation model and with inclusion of planarity restraints. Interestingly, MD simulations support the hypothesis that two binding sites for Na⁺ cations are localized close to the G-quartets inside the VK34 tetramer. The explicit water simulations are discussed in the methods section on p. 25 of the revised manuscript and the results are presented in Supplementary Figures 15 and 29.

Reviewer #2

In this study, Kocman and Plavec present NMR structures of a family of novel four-stranded nucleic acid structures formed in sequences with GCGAGGG repeats, which appear to be stabilized by GAGA- and GCGC-quartets. GAGA- and GCGC- quartets are distinct from the previously characterized Watson-Crick base pairs found in B-DNA or Hoogsteen G-quartets observed in G-quadruplexes. Their detailed characterization reveals many unique and surprising features which could not be predicted otherwise. The presented NMR data is extensive and the NMR structure determination is thorough in detail and of high quality. The approach taken is rational and straightforward. Notably, the authors determined the high resolution structures for multiple experimental DNA systems and provided additional

support these with mutational and CD data. These structures are distinctly different from the four-stranded DNA structures they previously reported in closely related sequences of GGGAGCG repeats, which are stabilized exclusively by non-canonical base pairing. The demonstration of GAGA- and GCGC-quartets as a stable structure motif broadens the known space in which non-canonical nucleic acid structures may be formed. Significantly, the authors found that sequences capable of forming such four-stranded DNA structures, which they named AGCGA-quadruplexes, exist in the regulatory regions of 38 human genes connected to multiple diseases, suggesting that such DNA structures may carry implications in disease. The novel and unique structure of these tetrahelical structures adds valuable contributions both to the literature-base of structural biology and to the general understanding of nucleic acid structures, and in my opinion, is appropriate for publication in Nature Communications.

I have some comments for the authors to consider.

1. All the sequences in this start and end with guanine. Have the authors tested the effect of flanking DNA?

We have added flanking adenine residues at the 5' (A_VK34), 3' (VK34_A) and both ends (A_VK34_A) of the VK34 oligonucleotide. Comparison of their respective 1D ¹H and 2D NOESY spectra with the parent spectra demonstrated that they all fold into very similar dimeric and tetrameric structures. The results demonstrating that flanking adenines minimally affect the dimeric and tetrameric VK34 topologies, but slightly influence their relative populations, are shown in Supplementary Figure 16 and commented on p. 10 of the revised manuscript.

2. As this structure contains a number of novel base-pairing schemes, I think it would be helpful to have schematic representations of all observed quartets and base-pairs as chemical structures including their hydrogen bonding.

A figure of schematic representation of G-G base pairs in N1-carbonyl symmetric geometry, G-A base pairs in N1-N7, carbonyl-amino geometry and minor as well as major groove GCGC-quartets was added as Figure 1 on page 3 of the revised manuscript.

3. On page 9, line 269-74, the description of the folding model for the formation of VK34 tetramer was unclear. In line 272, should "Fig. 4a and 4b" be "4a and 4d"?

The formation of the VK34 tetramer is illustrated more clearly in the new Figure 6b on p. 12 and is explained and cited on p. 11 of the revised manuscript.

4. On page 11, in the first paragraph, for the longer oligonucleotides 2VK34 and 4VK34, it would be helpful to include the 1D proton NMR spectra in addition to the 2D data shown in supplementary Fig. 18.

The 1D proton NMR spectra in addition to the 2D data are now shown for the longer oligonucleotides 2VK34 and 4VK34 in Figure 9 on p. 17 of the revised manuscript.

5. In supplementary Fig. 7, several imino protons are shown as doublets or multiplets. A discussion would be helpful.

The imino protons are highly sensitive to their chemical environments. The apparent doublets and multiplets for imino proton signals are not surprising since VK34 tetramer structure exhibits slight breaking of symmetry between its top and bottom parts. Additionally, slight chemical shift differences of imino proton signals are attributed to some structural 'breathing' between free G-C base pairs and GCGC-quartets. These observations and their interpretations are added to the Supplementary Figure 8.

6. Figure 6 is very intriguing and significant. The proposed intramolecular AGCGA-quadruplexes are mainly built upon the two core GCGC-quartets sandwiched by GAGA-quartets (shown in 6a and 6b left), which has been found in different human genes. This is different from the core of both VK34 dimer and tetramer. It would be very nice to see the molecular structure, or at least confirmation of the folding structure, of such an intramolecular AGCGA-quadruplex.

A unimolecular structure that has four G-C base pairs in the center with two G-A base pairs on each side has been observed and presented as VK2 in our earlier article (Nat. Commun. 5: 5831, 2014). During the current study we realized that monomeric structure adopted by VK2 is a member of a new tetrahelical family of intramolecular AGCGA-quadruplexes. This sub-family of AGCGA-quadruplexes is presented in Figure 10a of the revised manuscript.

Reviewers' Comments:

Reviewer #1:

Remarks to the Author:

The authors appear to have addressed or commented on all of the original reviews concerns. They have added new experimental results and documentation of their structures. This is a very nice body of work and increases our knowledge. Therefore, it is recommended for publication.

Reviewer #2:

Remarks to the Author:

The authors have addressed my questions and, in my opinion, the manuscript is appropriate for publication in Nature Communications.